# Exploring the Edge of Stability: Insights from a Fine-Grained Analysis of Gradient Descent in Shallow ReLU Networks

## Abstract

Gradient descent (GD) in modern neural networks initially sharpens the loss landscape by increasing the top Hessian eigenvalues until the step size becomes unstable. Subsequently, it enters the "Edge of Stability" (EoS) regime, characterized by unstable step size and non-monotonic loss reduction. EoS regime challenges conventional step size wisdom, sparking recent intensive research. However, a detailed characterization of EoS within the fine-grained GD neural network training dynamics remains under-explored. This paper provides a comprehensive analysis of both the sharpening phase and the EoS regime throughout the entire GD dynamics, focusing on shallow ReLU networks with squared loss on orthogonal inputs. Our theory characterizes the evolution of the top Hessian eigenvalues and elucidates the mechanisms behind EoS training. Leveraging this analysis, we present empirical validations of our predictions regarding sharpening and EoS dynamics, contributing to a deeper understanding of neural network training processes.

## 1 Introduction

Gradient descent (GD) has demonstrated remarkable effectiveness in training neural networks, yet our theoretical understanding of its operation remains incomplete. Conventional optimization theory relies on the descent lemma, which assures GD's monotonic loss reduction with a step size $\eta$, provided the maximum Hessian eigenvalue $\lambda_1$ remains below a critical sharpness threshold $\lambda^* = 2/\eta$. However, if $\lambda_1$ exceeds $\lambda^*$, GD iterations diverge. The descent lemma has formed the basis for practical heuristics and convergence theories associated with GD and its variants.

A recent seminal discovery by Cohen et al. (2021) has drawn considerable attention within the research community (See Section 1.1) due to its divergence from conventional optimization theory. This work reveals the "Progressive Sharpening" (PS) regime, in which GD systematically sharpens the loss landscape until the top Hessian eigenvalues reach the critical sharpness threshold $\lambda^*$. PS regime starkly contrasts with quadratic loss, which has globally constant Hessian eigenvalues. Subsequently, GD transitions into the "Edge of Stability" (EoS) regime. In the EoS, Hessian eigenvalues hover just above $\lambda^*$, and the loss undergoes a non-monotonic reduction accompanied by oscillations and sporadic spikes. In contrast, GD would diverge in the EoS regime for quadratic loss functions.

Understanding these intriguing phenomena within modern neural networks presents significant theoretical challenges due to the intricate interactions among various configurations involved in neural network training. Existing analyses often resort to simplified models that focus solely on EoS dynamics along restricted dimensions. While these analyses yield valuable insights, their direct applicability to neural networks remains somewhat elusive. Our investigations have unveiled a broader perspective: the PS and EoS regimes span multiple eigenspaces, each operating independently. (See Figure 1.) Furthermore, while the PS regime is typically simulated using normalization techniques that increase sharpness or step size as loss decreases, the PS regime can still occur empirically without such normalization. Our analysis demonstrates the PS regime without any normalization.

Our work embarks on a nuanced exploration, characterizing the PS and EoS regimes through a fine-grained analysis of GD dynamics in shallow ReLU networks. We extend and refine the insights from recent research by Boursier et al. (2022), which offers a comprehensive characterization of gradient flow (GF) dynamics for training two-layer ReLU networks with squared loss functions and orthog-

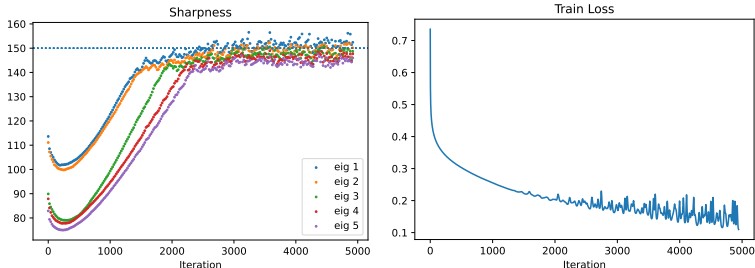

Figure 1: **Sharpness of the Top-5 Eigenvalues.** A fully-connected ReLU network is trained to completion using gradient descent on a 5k subset of the CIFAR-10 dataset. The left plots the sharpness of the top 5 eigenvalues during training. The progression of training loss is shown in the right figure.

onal inputs at small initialization. Our extension from GF to GD may be of independent interest due to unique challenges arising from the broken invariances and discretization errors. Leveraging our extensive characterization of GD dynamics, we elucidate the driving forces behind progressive sharpening in neural networks and the emergence of independent EoS phenomena across multiple eigenspaces.

## 1.1 RELATED WORKS

Our motivation stems from the empirical demonstration of the EoS phenomenon in Cohen et al. (2021). However, we note that similar empirical observations have surfaced in other works as well (Xing et al., 2018; Jastrzębski et al., 2020; Lewkowycz et al., 2020; Ahn et al., 2022b).

Recent years have witnessed a surge in research dedicated to unraveling the mysteries of the Progressive Sharpening (PS) and Edge of Stability (EoS) regimes in neural network dynamics. Given the formidable complexity of fully characterizing these dynamics, many studies have delved into minimalistic models, isolating specific aspects of the EoS phenomenon. For instance, Zhu et al. (2023) examined GD trajectories in a 2-parameter 4-layer 1-width scalar network, showcasing their convergence towards a minimum with EoS sharpness. Chen & Bruna (2023) explored fixed points for two-step updates in a 2-layer 1-width network and matrix factorization. Agarwala et al. (2023) scrutinized the quadratic regression model, establishing EoS behavior in two dimensions. Kreisler et al. (2023) focused on the linear scalar network, revealing that each GD update reduces the sharpness of the Gradient Flow (GF) solution initiated at the current point. Ahn et al. (2022a) provided insights into the bouncing dynamics of EoS in 2-dimensional loss functions and phase transitions in approximate GD dynamics when input weights are fixed in 2-layer ReLU networks. Building upon this foundation, Song & Yun (2023) extended and elucidated the trajectory alignment behavior of EoS dynamics within 2-dimensional loss landscapes through bifurcation theory. Lastly, Wu et al. (2023) showcased EoS dynamics in 2-layer diagonal linear networks. In contrast, our paper presents a comprehensive analysis of the full GD dynamics of 2-layer ReLU networks, where all parameters are trained.

Using minimalistic models simplifies the analyses of GD dynamics and sharpness, thereby facilitating the EoS phenomena in isolation. However, it is notable that conventional analyses predominantly concentrate on a singular EoS direction and its interaction with non-sharp direction, inadvertently overlooking the intricate interplay among multiple EoS directions that frequently emerge during neural network training. In this paper, we address this significant gap in the literature by conducting a comprehensive examination that accounts for the often neglected multi-directionality inherent in such scenarios. Our study delves into the evolution of not just one but multiple EoS directions, contributing to a more nuanced understanding of the complexities involved in network convergence.

Another avenue of research seeks to delineate the general conditions conducive to the emergence of the EoS phenomenon. Ma et al. (2022) demonstrated that sub-quadratic loss functions prevent the unstable GD from diverging. Arora et al. (2022) and Lyu et al. (2022) illustrated how, as loss

decreases, normalization amplifies the effective step size, ushering in the EoS regime. Wang et al. (2022) employed the output weight norm as a proxy to the sharpness, unveiling a four-phase cycle arising from the interplay between output weight norm and loss gradient. Damian et al. (2023) proposed a similar four-stage cycle but rooted in self-stabilizing mechanisms stemming from higher-order terms in Taylor expansions. Even et al. (2023) explored the influence of stochasticity and large step sizes on GD and stochastic GD in diagonal linear networks for overparametrized regression. Most recently, MacDonald et al. (2023) established connections between Hessian eigenvalues and Jacobian singular values. The identification of these general conditions where PS and EoS may occur does not imply that neural networks invariably exhibit PS and EoS due to these conditions. Our work adopts a distinct approach by scrutinizing the detailed GD dynamics of ReLU networks.

## 2 PRELIMINARIES

### 2.1 NOTATIONS AND SETUP

**Notations.** For $m \in \mathbb{N}$, we define $[m] = \{1, 2, \cdots, m\}$. We denote a vector by a lower case bold letter. For a vector $\boldsymbol{x}$, we represent its Euclidean norm as $\|\boldsymbol{x}\|$. We use $\mathbb{1}$ to represent the indicator function having value 1 when the condition in subscript is true and 0 otherwise. We use subscripts to represent index for neurons, data points or iterates. When there is a conflict, we put iterate index in superscripts.

**Shallow ReLU network.** We consider a two-layer ReLU network

$$h_{\boldsymbol{\theta}}(\boldsymbol{x}) = \sum_{j=1}^{m} a_j \sigma(\langle \boldsymbol{w}_j, \boldsymbol{x} \rangle),$$

where $m$ denotes the width of the network and the activation function $\sigma$ is the ReLU function. $\boldsymbol{w}_j \in \mathbb{R}^d$ and $a_j \in \mathbb{R}$ represents the input and output weights of the $j$-th neuron for $j \in [m]$. We encapsulate all parameters in $\boldsymbol{\theta} \in \mathbb{R}^{md+m}$.

**Mean squared error loss.** We use the mean squared error loss function

$$\mathcal{L}(\boldsymbol{\theta}) = \frac{1}{2n} \sum_{k=1}^{n} (h_{\boldsymbol{\theta}}(\boldsymbol{x}_k) - y_k)^2,$$

where $n$ denotes the number of data. $\boldsymbol{x}_k \in \mathbb{R}^d$ and $y_k \in \mathbb{R}$ represents the $k$-th input and output data point for $k \in [n]$. Following Boursier et al. (2022), we make the following assumptions on the data.

**Assumption 2.1.** *The input points are orthogonal to each other. Formally, we assume $\langle x_k, x_{k'} \rangle = \mathbb{1}_{k=k'}$ for all $k, k' \in [n]$. The output is nonzero $y_k \neq 0$ for all $k \in [n]$ and unbalanced: $\sum_{k|y_k>0} y_k^2 \neq \sum_{k|y_k<0} y_k^2$. Without loss of generality, we take $\sum_{k|y_k>0} y_k^2 > \sum_{k|y_k<0} y_k^2$.*

**Gradient descent.** We study the gradient descent (GD) with the update rule

$$\boldsymbol{\theta}_{t+1} - \boldsymbol{\theta}_t = -\eta \nabla \mathcal{L}(\boldsymbol{\theta}) = -\frac{\eta}{n} \sum_{k=1}^{n} (h_{\boldsymbol{\theta}}(\boldsymbol{x}_k) - y_k) \nabla h_{\boldsymbol{\theta}}(\boldsymbol{x}_k).$$

Here, $\eta$ is the step size. We introduce the error weighted data vector

$$\boldsymbol{d}_t = -\frac{1}{n} \sum_{k=1}^{n} \mathbb{1}_{\langle \boldsymbol{w}_j^t, \boldsymbol{x}_t \rangle > 0} (h_{\boldsymbol{\theta}}(\boldsymbol{x}_k) - y_k) \boldsymbol{x}_k$$

to succinctly write the dynamics as

$$\boldsymbol{w}_j^{t+1} - \boldsymbol{w}_j^t = \eta a_j^t \boldsymbol{d}_j^t \quad \text{and} \quad a_j^{t+1} - a_j^t = \eta \langle \boldsymbol{w}_j^t, \boldsymbol{d}_j^t \rangle. \tag{1}$$

We initialize our parameters as

$$\boldsymbol{w}_j^0 = \lambda \boldsymbol{g}_j \quad \text{and} \quad a_j^0 = \lambda s_j,$$

where $\boldsymbol{g}_j$ and $s_j$ are drawn from the uniform distributions $U(\mathbb{S}^{d-1})$ and $U(\{-1, 1\})$, respectively. The parameter $0 < \lambda < 1$ controls the magnitude at initialization.

## 2.2 BASIC PROPERTIES

Now we state lemmas for basic properties of GD dynamics in our settings.

**Lemma 2.1.** *(Approximate Balance) GD iterates slowly update the balance:*

$$\left(\|\boldsymbol{w}_j^{t+1}\|^2 - |a_j^{t+1}|^2\right) - \left(\|\boldsymbol{w}_j^t\|^2 - |a_j^t|^2\right) = \eta^2 \left(\|\boldsymbol{d}_j^t\|^2 |a_j^t|^2 - \langle \boldsymbol{d}_j^t, \boldsymbol{w}_j^t \rangle^2\right). \tag{2}$$

*Equivalently, the balance updates as*

$$\|\boldsymbol{w}_j^{t+1}\|^2 - |a_j^{t+1}|^2 = \left(1 - \eta^2 \|\boldsymbol{d}_j^t\|^2\right) \left(\|\boldsymbol{w}_j^t\|^2 - |a_j^t|^2\right) + \eta^2 \left(\|\boldsymbol{d}_j^t\|^2 |a_j^t|^2 - \langle \boldsymbol{d}_j^t, \boldsymbol{w}_j^t \rangle^2\right). \tag{3}$$

Since $|\langle \boldsymbol{d}_j^t, \boldsymbol{w}_j^t \rangle| \le \|\boldsymbol{d}_j^t\|\|\boldsymbol{w}_j^t\|$, if $\|\boldsymbol{w}_j^t\| \ge |a_j^t|$, then from the Equation 2, we have $\|\boldsymbol{w}_j^{t+1}\| \ge |a_j^{t+1}|$. If we initialize $\|\boldsymbol{w}_j^0\| \ge |a_j^0|$, then we have $\|\boldsymbol{w}_j^t\| \ge |a_j^t|$ for all $t \ge 0$. Note that the gradient flow would keep the balance unchanged (Arora et al., 2019). Our analysis initialize at balance i.e. $\|\boldsymbol{w}_j^0\| = |a_j^0|$ and bound the approximate balance throughout GD dynamics.

**Lemma 2.2.** *(Neuron Preactivation) GD iterates update the pre-activation value as*

$$\langle \boldsymbol{w}_j^{t+1}, \boldsymbol{x}_k \rangle - \langle \boldsymbol{w}_j^t, \boldsymbol{x}_k \rangle = \eta a_j^t \langle \boldsymbol{d}_j^T, \boldsymbol{x}_k \rangle = -\eta a_j^t \frac{1}{n} \mathbb{1}_{\langle \boldsymbol{w}_j^t, \boldsymbol{x}_k \rangle > 0} \left(h_{\boldsymbol{\theta}^t}(\boldsymbol{x}_k) - y_k\right).$$

If the neuron is inactive at some data as $\langle \boldsymbol{w}_j^t, \boldsymbol{x}_k \rangle \le 0$, then we have $\langle \boldsymbol{w}_j^{t+1}, \boldsymbol{x}_k \rangle = \langle \boldsymbol{w}_j^t, \boldsymbol{x}_k \rangle$. Thus, if the neuron is inactive at some data point at initialization as $\langle \boldsymbol{w}_j^0, \boldsymbol{x}_k \rangle \le 0$, then it will keep being inactive as $\langle \boldsymbol{w}_j^t, \boldsymbol{x}_k \rangle \le 0$ for all $t \ge 0$. Because of this property, we assume that some neurons at the beginning to be aligned with all data point. We refer to Boursier et al. (2022) for the justification of this assumption.

**Assumption 2.2.** *Define*

$$\mathbb{S}_{1,+} := \left\{ j \in [m] : s_j = +1 \text{ and for all } k \text{ such that } y_k > 0, \langle \boldsymbol{w}_j^k, \boldsymbol{x}_k \rangle \ge 0 \right\},$$

$$\mathbb{S}_{1,-} := \left\{ j \in [m] : s_j = -1 \text{ and for all } k \text{ such that } y_k < 0, \langle \boldsymbol{w}_j^k, \boldsymbol{x}_k \rangle \ge 0 \right\}.$$

*We assume that the sets $\mathbb{S}_{+,1}$ and $\mathbb{S}_{-,1}$ are both non-empty.*

## 3 MAIN RESULT

We state our main theorem.

**Theorem 3.1.** *Let $\boldsymbol{\theta} \in \mathbb{R}^{md+m}$ be ordered as*

$$\boldsymbol{\theta}_{(j-1)(d+1)+1:j(d+1)} = (\boldsymbol{w}_j; a_j).$$

*Define the vector $\boldsymbol{v}_+, \boldsymbol{v}_- \in \mathbb{R}^{md+m}$ as*

$$\boldsymbol{v}_{+,(j-1)(d+1)+1:j(d+1)} = \begin{cases} (\boldsymbol{d}_+, \|\boldsymbol{d}_+\|) & \text{if } j \in \mathbb{S}_{1,+}, \\ \boldsymbol{0}_d & \text{otherwise}, \end{cases}$$

*and*

$$\boldsymbol{v}_{-,(j-1)(d+1)+1:j(d+1)} = \begin{cases} (\boldsymbol{d}_-, \|\boldsymbol{d}_-\|) & \text{if } j \in \mathbb{S}_{1,-}, \\ \boldsymbol{0}_d & \text{otherwise}. \end{cases}$$

*Furthermore, define sharpness along these directions as*

$$\lambda_+ = \frac{\langle \boldsymbol{v}_+, \nabla^2 \mathcal{L}(\boldsymbol{\theta}) \boldsymbol{v}_+ \rangle}{\|vv_+\|^2}$$

*and*

$$\lambda_- = \frac{\langle \boldsymbol{v}_-, \nabla^2 \mathcal{L}(\boldsymbol{\theta}) \boldsymbol{v}_- \rangle}{\|vv_-\|^2}.$$

*Then, they develop in each phase as follows:*

- *Phase 1: The sharpness stays small throughout as*

$$\lambda_+ \leq 2|\mathbb{S}_{1,+}|\lambda^{1-r\epsilon} \text{ and}$$
$$\lambda_- \leq 2|\mathbb{S}_{1,-}|\lambda^{1-r\epsilon}.$$

- *Phase 2: The sharpness in positive direction grows rapidly as*

$$\lambda_+ \geq \frac{n\|\boldsymbol{d}_+\|^2}{4|\mathbb{S}_{1,+}|} \frac{1}{1 + \exp(-a_+(\lambda)(t - \tau))}$$

  *while the sharpness in negative direction stays small as*

$$\lambda_- \leq 2|\mathbb{S}_{1,-}|\lambda^{\epsilon}.$$

- *Phase 3: The sharpness in negative direction grows rapidly as*

$$\lambda_- \geq \frac{n\|\boldsymbol{d}_-\|^2}{4|\mathbb{S}_{1,-}|} \frac{1}{1 + \exp(-a_-(\lambda)(t - \tau))}.$$

Therefore, the smoothness along the two dimensions develops independently. Moreover, two dimensions have different growth rates of $a_+(\lambda)$ and $a_-(\lambda)$.

## 4 EXPERIMENTS

In this section, we verify our theory by tracking the trajectories of lower eigenvalues. A two-layer fully-connected ReLU network is trained with mean squared error (MSE) on a 5k subset of CIFAR-10 (Krizhevsky et al., 2009). The details of our experimental settings can be found in Appendix B.

As depicted in Figure 1, upon the entry of the maximum eigenvalue into the EoS phase, the stability of gradient descent is compromised. A noticeable consequence of this shift is the oscillation observed in the sharpness of the maximum eigenvalue, leading to corresponding oscillations in the remaining eigenvalues, all converging toward the threshold $2/\eta$. Intriguingly, we observe that the sharpness of each individual eigenvalue demonstrates independent movement in various directions during this process. Variations in directions are shown more distinctly in Figure 2. In every direction, there exists a distinct rate at which sharpness increases.

Further, we investigate the growth rate of sharpness of different eigenvalues through learning rate decay. We run gradient descent on the network with stepsize $\eta$ until the eigenvalues reach the threshold. The learning rate is reduced once the chosen number of eigenvalues is in the EoS phase. We vary the iteration step at which the learning rate decay occurs and compare the resulting figures (Figure 2). Regardless of the chosen steps, the growth rate of sharpness after learning rate decay displays a relatively constant progression. Observed dynamics of sharpness of eigenvalues verify the assumptions of our theory.

We perform additional experiments on the effects of varying widths, depths, activation functions, and loss functions on the evolution of sharpness. See Appendix C for details.

## 5 CONCLUSION

In this paper, we study the underlying mechanisms driving progressive sharpening and independent EoS behaviors across various eigenspaces. We propose a comprehensive exploration of GD dynamics in shallow ReLU networks, where all parameters are trained. Our theoretical study is evaluated via experiments on fully-connected ReLU networks. We analyze the trajectories of sharpness of lower eigenvalues and independent progression with different directions, supporting our theoretical analysis. Our work extends the understanding of the PS and EoS regimes to shallow ReLU networks, shedding light on the dynamics of GD in these regimes, even without normalization techniques.

## REFERENCES

Atish Agarwala, Fabian Pedregosa, and Jeffrey Pennington. Second-order regression models exhibit progressive sharpening to the edge of stability. In Andreas Krause, Emma Brunskill, Kyunghyun

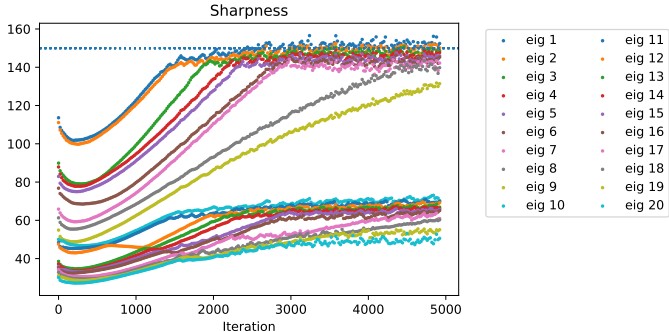

Figure 2: **Independent EoS Directions.** The evolution of the top 20 eigenvalues during the training of the fully-connected ReLU network is shown. The maximum eigenvalue reaches the EoS first, followed by the rest of the eigenvalues. Observe independent and diverse trajectories of the eigenvalues.

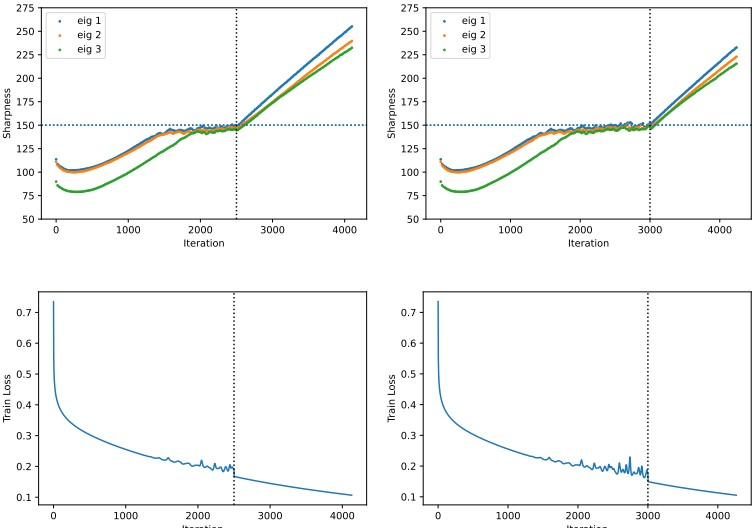

Figure 3: **Learning Rate Decay.** We observe the sharpness and training loss change due to learning rate decay. When the network is in EoS phase, the learning rate is reduced to $\eta/4$. A black vertical line shows the step at which learning rate decay occurs.

Cho, Barbara Engelhardt, Sivan Sabato, and Jonathan Scarlett (eds.), *Proceedings of the 40th International Conference on Machine Learning*, volume 202 of *Proceedings of Machine Learning Research*, pp. 169–195. PMLR, 23–29 Jul 2023. URL `https://proceedings.mlr.press/v202/agarwala23b.html`.

Kwangjun Ahn, Sébastien Bubeck, Sinho Chewi, Yin Tat Lee, Felipe Suarez, and Yi Zhang. Learning threshold neurons via the" edge of stability". *arXiv preprint arXiv:2212.07469*, 2022a.

Kwangjun Ahn, Jingzhao Zhang, and Suvrit Sra. Understanding the unstable convergence of gradient descent. In Kamalika Chaudhuri, Stefanie Jegelka, Le Song, Csaba Szepesvari, Gang Niu, and Sivan Sabato (eds.), *Proceedings of the 39th International Conference on Machine Learning*, volume 162 of *Proceedings of Machine Learning Research*, pp. 247–257. PMLR, 17–23 Jul 2022b. URL `https://proceedings.mlr.press/v162/ahn22a.html`.

Sanjeev Arora, Simon Du, Wei Hu, Zhiyuan Li, and Ruosong Wang. Fine-grained analysis of optimization and generalization for overparameterized two-layer neural networks. In Kamalika

Chaudhuri and Ruslan Salakhutdinov (eds.), *Proceedings of the 36th International Conference on Machine Learning*, volume 97 of *Proceedings of Machine Learning Research*, pp. 322–332. PMLR, 09–15 Jun 2019. URL `https://proceedings.mlr.press/v97/arora19a.html`.

Sanjeev Arora, Zhiyuan Li, and Abhishek Panigrahi. Understanding gradient descent on the edge of stability in deep learning. In Kamalika Chaudhuri, Stefanie Jegelka, Le Song, Csaba Szepesvari, Gang Niu, and Sivan Sabato (eds.), *Proceedings of the 39th International Conference on Machine Learning*, volume 162 of *Proceedings of Machine Learning Research*, pp. 948–1024. PMLR, 17–23 Jul 2022. URL `https://proceedings.mlr.press/v162/arora22a.html`.

Etienne Boursier, Loucas PILLAUD-VIVIEN, and Nicolas Flammarion. Gradient flow dynamics of shallow relu networks for square loss and orthogonal inputs. In S. Koyejo, S. Mohamed, A. Agarwal, D. Belgrave, K. Cho, and A. Oh (eds.), *Advances in Neural Information Processing Systems*, volume 35, pp. 20105–20118. Curran Associates, Inc., 2022. URL `https://proceedings.neurips.cc/paper_files/paper/2022/file/7eeb9af3eb1f48e29c05e8dd3342b286-Paper-Conference.pdf`.

Lei Chen and Joan Bruna. Beyond the edge of stability via two-step gradient updates. In Andreas Krause, Emma Brunskill, Kyunghyun Cho, Barbara Engelhardt, Sivan Sabato, and Jonathan Scarlett (eds.), *Proceedings of the 40th International Conference on Machine Learning*, volume 202 of *Proceedings of Machine Learning Research*, pp. 4330–4391. PMLR, 23–29 Jul 2023. URL `https://proceedings.mlr.press/v202/chen23b.html`.

Jeremy Cohen, Simran Kaur, Yuanzhi Li, J Zico Kolter, and Ameet Talwalkar. Gradient descent on neural networks typically occurs at the edge of stability. In *International Conference on Learning Representations*, 2021. URL `https://openreview.net/forum?id=jh-rTtvkGeM`.

Alex Damian, Eshaan Nichani, and Jason D. Lee. Self-stabilization: The implicit bias of gradient descent at the edge of stability. In *The Eleventh International Conference on Learning Representations*, 2023. URL `https://openreview.net/forum?id=nhKHA59gXz`.

Mathieu Even, Scott Pesme, Suriya Gunasekar, and Nicolas Flammarion. (s) gd over diagonal linear networks: Implicit regularisation, large stepsizes and edge of stability. *arXiv preprint arXiv:2302.08982*, 2023.

Stanisław Jastrzębski, Maciej Szymczak, Stanislav Fort, Devansh Arpit, Jacek Tabor, Kyunghyun Cho*, and Krzysztof Geras*. The break-even point on optimization trajectories of deep neural networks. In *International Conference on Learning Representations*, 2020. URL `https://openreview.net/forum?id=r1g87C4KwB`.

Itai Kreisler, Mor Shpigel Nacson, Daniel Soudry, and Yair Carmon. Gradient descent monotonically decreases the sharpness of gradient flow solutions in scalar networks and beyond. In Andreas Krause, Emma Brunskill, Kyunghyun Cho, Barbara Engelhardt, Sivan Sabato, and Jonathan Scarlett (eds.), *Proceedings of the 40th International Conference on Machine Learning*, volume 202 of *Proceedings of Machine Learning Research*, pp. 17684–17744. PMLR, 23–29 Jul 2023. URL `https://proceedings.mlr.press/v202/kreisler23a.html`.

Alex Krizhevsky, Geoffrey Hinton, et al. Learning multiple layers of features from tiny images. 2009.

Aitor Lewkowycz, Yasaman Bahri, Ethan Dyer, Jascha Sohl-Dickstein, and Guy Gur-Ari. The large learning rate phase of deep learning: the catapult mechanism. *arXiv preprint arXiv:2003.02218*, 2020.

Kaifeng Lyu, Zhiyuan Li, and Sanjeev Arora. Understanding the generalization benefit of normalization layers: Sharpness reduction. In S. Koyejo, S. Mohamed, A. Agarwal, D. Belgrave, K. Cho, and A. Oh (eds.), *Advances in Neural Information Processing Systems*, volume 35, pp. 34689–34708. Curran Associates, Inc., 2022. URL `https://proceedings.neurips.cc/paper_files/paper/2022/file/dffd1c523512e557f4e75e8309049213-Paper-Conference.pdf`.

Chao Ma, Daniel Kunin, Lei Wu, and Lexing Ying. Beyond the quadratic approximation: The multiscale structure of neural network loss landscapes. *Journal of Machine Learning*, 1(3): 247–267, 2022. ISSN 2790-2048. doi: https://doi.org/10.4208/jml.220404. URL `http://global-sci.org/intro/article_detail/jml/21028.html`.

Lachlan Ewen MacDonald, Jack Valmadre, and Simon Lucey. On progressive sharpening, flat minima and generalisation. *arXiv preprint arXiv:2305.14683*, 2023.

Minhak Song and Chulhee Yun. Trajectory alignment: Understanding the edge of stability phenomenon via bifurcation theory. *arXiv preprint arXiv:2307.04204*, 2023.

Zixuan Wang, Zhouzi Li, and Jian Li. Analyzing sharpness along gd trajectory: Progressive sharpening and edge of stability. In S. Koyejo, S. Mohamed, A. Agarwal, D. Belgrave, K. Cho, and A. Oh (eds.), *Advances in Neural Information Processing Systems*, volume 35, pp. 9983–9994. Curran Associates, Inc., 2022. URL `https://proceedings.neurips.cc/paper_files/paper/2022/file/40bb79c081828bebdc39d65a82367246-Paper-Conference.pdf`.

Jingfeng Wu, Vladimir Braverman, and Jason D Lee. Implicit bias of gradient descent for logistic regression at the edge of stability. *arXiv preprint arXiv:2305.11788*, 2023.

Chen Xing, Devansh Arpit, Christos Tsirigotis, and Yoshua Bengio. A walk with sgd. *arXiv preprint arXiv:1802.08770*, 2018.

Xingyu Zhu, Zixuan Wang, Xiang Wang, Mo Zhou, and Rong Ge. Understanding edge-of-stability training dynamics with a minimalist example. In *The Eleventh International Conference on Learning Representations*, 2023. URL `https://openreview.net/forum?id=p7EagBsMAEO`.

## A    MAIN PROOFS

### A.1    PHASE 1

During phase 1, GD aligns neurons with the data while the weights remain small in magnitude and input-output layers remain balanced. Concretely, for $t$-th iteration in phase 1, the neurons remains small as

$$\|\boldsymbol{w}_j^t\| \leq A\lambda, |a_j^t| \leq A\lambda,$$

for some $A > 1$; the balance is approximately maintained as

$$\|\boldsymbol{w}_j^t\| - |a_j^t| \leq B\lambda$$

for some $B < 1$; and the output weight does not change the sign as

$$\text{sign}(a_j^t) = \text{sign}(a_j^0).$$

The alignment that is achieved at the end of the training is represented as

$$\langle \hat{\boldsymbol{w}}_j^{\tau_1}, \boldsymbol{d}_+ \rangle \geq (1 - factor)\|\boldsymbol{d}_+\|$$

for $j \in S_{1,+}$ and

$$\langle \hat{\boldsymbol{w}}_j^{\tau_1}, \boldsymbol{d}_- \rangle \geq (1 - factor)\|\boldsymbol{d}_-\|$$

for $j \in S_{1,-}$.

We begin our proof by defining the time period during which the weights remain small in magnitude and input-output layers remain balanced.

**Lemma A.1.** *Let $A > B + 1 > 1$. Define*

$$\tau_{init}(A, B) = \inf\{t \geq 0 : \exists j \in [m] \text{ such that either } \|\boldsymbol{w}_j^t\| > A\lambda \text{ or } |a_j^t| > A\lambda \text{ or }$$

$$\|\boldsymbol{w}_j^t\| - |a_j^t| > B\lambda \text{ or } \text{sign}(a_j^t) \neq \text{sign}(a_j^0)\}. \quad (4)$$

*Denote $L = \|D_+\| + mA^2\lambda^2$. Then, GD with step size $\eta < \frac{1}{L}$ satisfies*

$$\tau_{init}(A, B) \geq \max\left\{\frac{1}{\eta L}\log\frac{A}{1+B}, \frac{1}{\eta L}\log\frac{B+1}{B}, \frac{1}{(\eta L)^2}\log\frac{A^2}{A^2 - B^2}\right\}.$$

*Proof.* Let $t \leq \tau_{init} := \tau_{init}(A, B)$. First, for $k \in [m]$, we bound the output as

$$|h_t(\boldsymbol{x}_k)| \leq \sum_{j=1}^{m} |a_j^t| |\sigma(\langle \boldsymbol{w}_j^t, \boldsymbol{x}_k \rangle)| \leq \sum_{j=1}^{m} |a_j^t| \|\boldsymbol{w}_j^t\| \leq mA^2\lambda^2.$$

Thus, from the assumption $mA^2\lambda^2 \leq \min_k |y_k|$, during $t \leq \tau_{init}$, the error $h_t(\boldsymbol{x}_k) - y_k$ and the label $y_k$ have the opposite sign.

The output neuron update has the form

$$a_j^{t+1} - a_j^t = \eta \langle \boldsymbol{d}_j^t, \boldsymbol{w}_j^t \rangle = -\frac{\eta}{n} \sum_{k=1}^{n} \mathbb{1}_{\langle \boldsymbol{w}_j^t, \boldsymbol{x}_k \rangle > 0} (h_t(\boldsymbol{x}_k) - y_k) \langle \boldsymbol{x}_k, \boldsymbol{w}_j^t \rangle.$$

Define the error weighted data vectors

$$\boldsymbol{d}_{j,+}^t = -\frac{1}{n} \sum_{k=1}^{n} \mathbb{1}_{\langle \boldsymbol{w}_j^t, \boldsymbol{x}_k \rangle > 0} \mathbb{1}_{y_k > 0} (h_t(\boldsymbol{x}_k) - y_k) \boldsymbol{x}_k,$$

$$\boldsymbol{d}_{j,-}^t = -\frac{1}{n} \sum_{k=1}^{n} \mathbb{1}_{\langle \boldsymbol{w}_j^t, \boldsymbol{x}_k \rangle > 0} \mathbb{1}_{y_k < 0} (h_t(\boldsymbol{x}_k) - y_k) \boldsymbol{x}_k.$$

Then, we have

$$\boldsymbol{d}_j^t = \boldsymbol{d}_{j,+}^t + \boldsymbol{d}_{j,-}^t, \langle \boldsymbol{d}_{j,+}^t, \boldsymbol{w}_j^t \rangle > 0, \langle \boldsymbol{d}_{j,-}^t, \boldsymbol{w}_j^t \rangle < 0,$$

$$\|\boldsymbol{d}_{j,+}^t\| \leq \|\boldsymbol{d}_+\| + mA^2\lambda^2 \text{ and } \|\boldsymbol{d}_{j,-}^t\| \leq \|\boldsymbol{d}_-\| + mA^2\lambda^2.$$

We recall the definition of data vectors

$$\boldsymbol{d}_+ = \frac{1}{n} \sum_{k=1}^{n} \mathbb{1}_{y_k > 0} y_k \boldsymbol{x}_k,$$

$$\boldsymbol{d}_- = \frac{1}{n} \sum_{k=1}^{n} \mathbb{1}_{y_k < 0} y_k \boldsymbol{x}_k$$

and the assumption $\|\boldsymbol{d}_+\| > \|\boldsymbol{d}_-\|$. Using these notations, the output neuron update is bounded as

$$a_j^{t+1} - a_j^t \leq \eta \langle \boldsymbol{d}_{j,+}^t, \boldsymbol{w}_j^t \rangle \leq \eta \|\boldsymbol{d}_{j,+}^t\| \|\boldsymbol{w}_j^t\| \leq \eta \left( \|\boldsymbol{d}_+\| + mA^2\lambda^2 \right) \left( |a_j^t| + B\lambda \right),$$

$$a_j^{t+1} - a_j^t \geq \eta \langle \boldsymbol{d}_{j,-}^t, \boldsymbol{w}_j^t \rangle \geq \eta \|\boldsymbol{d}_{j,-}^t\| \|\boldsymbol{w}_j^t\| \geq -\eta \left( \|\boldsymbol{d}_-\| + mA^2\lambda^2 \right) \left( |a_j^t| + B\lambda \right)$$

and consequently

$$\left| |a_j^{t+1}| - |a_j^t| \right| \leq |a_j^{t+1} - a_j^t| \leq \eta \left( \|\boldsymbol{d}_+\| + mA^2\lambda^2 \right) \left( |a_j^t| + B\lambda \right). \tag{5}$$

Applying Lemma A.2, we obtain

$$|a_j^t| \leq (1 + \eta L)^t (|a_j^0| + B\lambda) - B\lambda,$$

$$\|\boldsymbol{w}_j^t\| \leq (1 + \eta L)^t (|a_j^0| + B\lambda).$$

From approximate balancedness in Lemma 2.1 $\qquad \square$

## A.2 AUXILIARY LEMMAS

This section lists and proves the lemma used throughout our main proofs.

**Lemma A.2.** *Let $A > 0$ and $B = AC - A$. If a sequence $\{a_t\}_{t \geq 0}$ is recursively bounded as*

$$a_{t+1} \leq Aa_t + B$$

*for each $t \geq 0$, then we have a bound on the sequence as*

$$a_t \leq A^t(a_0 + C) - C.$$

*If $a_{t+1} \geq Aa_t + B$, then $a_t \geq A^t(a_0 + C) - C$.*

*Proof.* The lemma follows from the simple observation

$$a_{t+1} + C \leq A(a_t + C).$$

The other inequality can be proven similarly. $\qquad \square$

# B EXPERIMENTAL SETTINGS

**Training data:** Our hypothesis is experimented on a subset of the CIFAR-10 image classification dataset (Krizhevsky et al., 2009). The dataset consists of 60,000 32x32 color images in 10 classes, with 6000 images per class. We sampled 5,000 images for our experiments.

**Architectures:** We examine a fully-connected network with an activation function, tanh or ReLU.

The PyTorch code for the fully-connected tanh network is provided as follows:

```
nn.Sequential(
    nn.Flatten(start_dim=1, end_dim=-1),
    nn.Linear(3072, 128, bias=True),
    nn.Tanh(),
    nn.Linear(128, 1, bias=True)
)
```

For the fully-connected ReLU network, the tanh function is substituted with ReLU function:

```
nn.Sequential(
    nn.Flatten(start_dim=1, end_dim=-1),
    nn.Linear(3072, 128, bias=True),
    nn.ReLU(),
    nn.Linear(128, 1, bias=True)
)
```

With the above networks as base models, we experiment on fully-connected networks with different network widths, depths, and loss functions. Networks are trained until 99% accuracy is reached. A stepsize $\eta$ is chosen from one of the subsets $\{2/200, 2/150, 2/50, 2/30\}$.

# C EXPERIMENTAL RESULTS

In this section, we observe the evolution of the sharpness of training fully-connected networks when their widths, depths, and loss functions are varied.

**Sharpness of Lower Eigenvalues:** We observe the progression of top-15 eigenvalues when the fully-connected (FC) tanh network is trained until convergence. As shown in Figure 4, once the maximum eigenvalue crosses $2/\eta$, all eigenvalues exhibit a synchronized destabilization, subsequently embarking on a discernible ascent, gradually converging toward the value $2/\eta$. The eigenvalues of the FC tanh network also display different trajectories in many directions.

**Learning Rate Decay:** In our comprehensive analysis, we focus on the discernible effects of learning rate decay on the sharpness and training loss dynamics within the neural network. Specifically, during the EoS (Edge of Stability) phase of network training, a strategic reduction of the learning rate to $\eta/4$ is implemented. As shown in Figure 5, this pivotal adjustment instigates a profound transformation in the network's behavior. Notably, following the application of learning rate decay, the evolution of sharpness undergoes a stark alteration. The previously observed oscillations during the EoS phase come to a halt, indicating a newfound stability in the training process. Furthermore, in terms of Mean Squared Error (MSE), we observe a pronounced and monotonic increase in sharpness, while in the case of Cross-Entropy (CE), a smooth and logarithmic-like progression is evident. Equally noteworthy is the cessation of oscillations in the training loss, which subsequently embarks on a consistent and monotonic decline.

**Effects of Network Size on Shaprness:** In this section, we explore the relationship between network architecture and sharpness. Figures 6 and 7 reveal insights into the effects of varying the network's depth and width. When network widths vary, shallower network architectures tend to reach the Edge of Stability (EoS) phase at an earlier stage in their training trajectory, demonstrating an accelerated entry into this pivotal state. However, it is noteworthy that these shallower networks

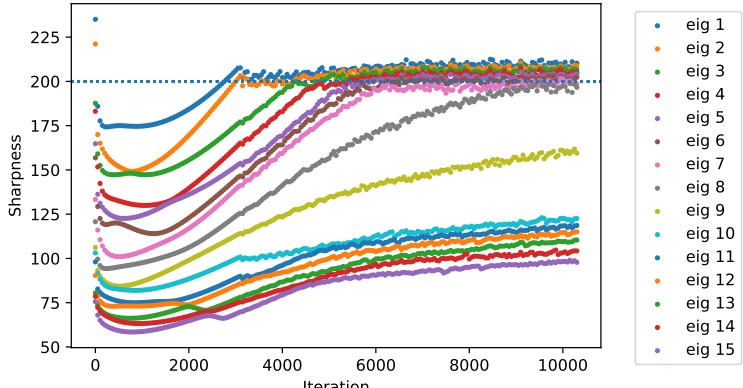

Figure 4: **Progression of 15 Eigenvalues.** The plot depicts different trajectories of 15 eigenvalues of the fully-connected tanh network. When trained to completion, the first eigenvalue enters the EoS phase first, and the rest follows in turn.

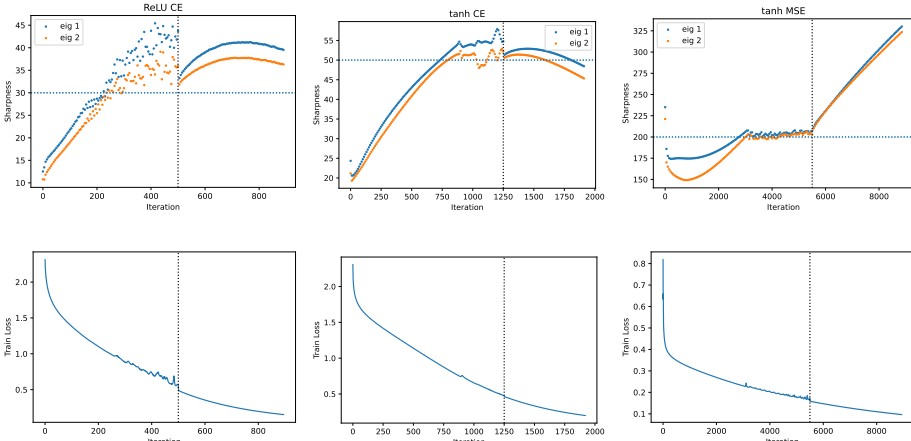

Figure 5: **Learning Rate Decay.** We observe the sharpness and training loss change due to learning rate decay. When the network is in EoS phase, the learning rate is reduced to $\eta/4$. A black vertical line shows the step at which learning rate decay occurs.

linger within the EoS phase for a more extended period. Networks display similar behavior when depths are varied. The deeper the network is, the more gradual the progression of sharpness is throughout the training process.

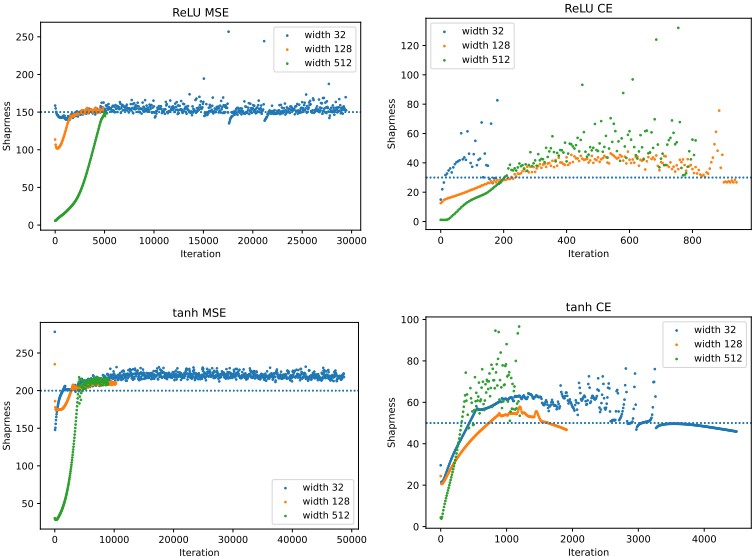

Figure 6: **Width Variations.** We track the evolution of sharpness during the training of fully-connected networks with varying widths. The depth-1 network is the standard architecture. Shallower networks reach the EoS phase earlier but stay longer.

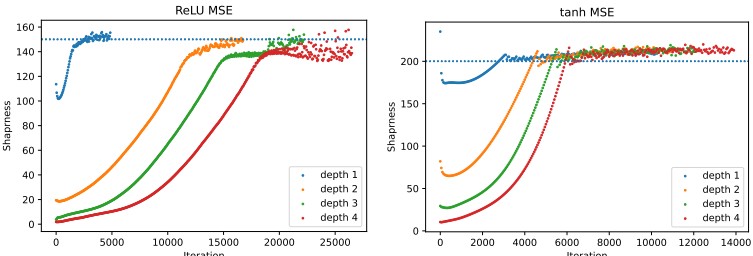

Figure 7: **Depth Variations.** We track the evolution of sharpness during the training of fully-connected networks with varying depths. The depth-1 network is the standard architecture. Deeper networks show slower progression of sharpness during training.

