# OpenReview forum: "Exploring the Edge of Stability: Insights from a Fine-Grained Analysis of Gradient Descent in Shallow ReLU Networks"
_ICLR.cc/2024/Conference — ICLR 2024 Conference Withdrawn Submission_

### Official Review · Reviewer_CLYU · 2023-10-19

**Soundness:** 1 poor
**Presentation:** 2 fair
**Contribution:** 1 poor
**Rating:** 3
**Confidence:** 4

**Summary:**

This paper provides an analysis for shallow ReLU networks with squared loss on orthogonal inputs, with a particular focus on the dynamics of a certain measure of sharpness. The paper also argues that the top eigenvalues can be characterized by the theory part, indicating the Progressive sharpening (PS) and Edge-of-Stability (EoS) regimes span multiple independent eigenspaces.

**Strengths:**

The strength of this work lies in its reliance on an approachable data assumption, specifically orthogonal data. Furthermore, the authors make an effort to comprehend the entire top eigenspace, rather than solely focusing on the largest one.

**Weaknesses:**

However, the limitations in the theoretical and empirical contributions of this paper.

For the main theorem.
- Firstly, the presentation of the primary result may benefit from improved clarity. There is a need for further definition and explanation of notations such as $vv+$, $vv-$, $r$, and $\tau$. These variables appear undefined in the paper.
- What is the main message for the main theorem? It requires more explicit elucidation of the intuition behind the significance of $v+/v-$ and its connection to the top eigenspace. Also, please write some proof intuition.
- In its current form, there is no connection between the main theorem and the PS or EoS phenomenon. Especially for the EoS part: the main theorem seems to imply certain directional sharpness is growing. But the key characteristic of the EoS regime is the oscillation for the top eigenvalue. Please justify the relation between the main theorem and the EoS phenomenon.

For the experiment:
- I don't think the experiments can be a verification for the theorem. The main theorem is only about two directions, and has nothing to do with the oscillation. The experiments cannot corroborate any of the assumptions: CIFAR-10 is not orthogonal dataset, and the training plot cannot indicate Assumption 2. I don't know why the authors claim "Observed dynamics of sharpness of eigenvalues verify the assumptions of our theory." Please elaborate the connection between the experiment and the theoretical results.
- Also, I don't think the multiple unstable eigenvalue view is new. In Damian et al. 2023, Section 8.2 already discussed this phenomenon. Furthermore, the top eigenspaces (top 1-20) are obviously not **independent**, since when the largest eigenvalue exceed the stability threshold, all small eigenvalues (below the stability threshold) also oscillates.

Overall, I have issues with the presentation of the main theorem and its true correlation with the EoS or PS phenomenon. Some of the claim of this paper in the experiments is incorrect.

**Questions:**

Please address the questions in the weaknesses section.

---

### Official Review · Reviewer_xPNL · 2023-10-23

**Soundness:** 2 fair
**Presentation:** 2 fair
**Contribution:** 1 poor
**Rating:** 3
**Confidence:** 5

**Summary:**

The paper under review provides an analysis of the progressive sharpening and edge of stability phenomena in the context of training two-layer ReLU networks.  Particularly, the authors explore the dynamics of sharpness along two designated directions.

**Strengths:**

- The paper's main contribution lies in its demonstration that, under certain conditions, the sharpness along specific directions will increase during particular phases of training in a two-layer ReLU network.
- The experiments also confirm that sharpness along most eigen-directions will increase monotonically.

**Weaknesses:**

The analysis is still rather preliminary.

- The specfic directions in Theorem 3.1 are strange and it is unclear how they are related to eigen-directions, which ones are analyzed in the experiments.


- The authors provide only a lower bound for the sharpness along the two directions they selected. They fail to establish whether this sharpness will cease to grow upon reaching the well-known "edge of stability" upper bound $2/\eta$. Additionally, the paper lacks an upper bound for the increasing sharpness, leaving its relation to the edge of stability ambiguous.



- The paper's main theoretical claim that the sharpness will increase at varying rates in different phases does not find empirical support in the conducted experiments, where the eigenvalues seem to increase almost concurrently.


**Minor Points:**
This paper credits only Cohen et al. (2021) for the "discovery" of the edge of stability and progressive sharpening. This significantly overlooks previous works in this field. Specifically:
- The edge of stability was first observed for convergent GD solutions by Wu et al. (2018)—a point acknowledged by Cohen et al. (2021).
- The phenomenon of progressive sharpening was originally discovered by Jastrzebski et al. (2020), which builds on the work of Wu et al. (2018).

**Questions:**

None

---

### Official Review · Reviewer_5wmW · 2023-10-29

**Soundness:** 1 poor
**Presentation:** 1 poor
**Contribution:** 1 poor
**Rating:** 3
**Confidence:** 4

**Summary:**

I found this paper submission quite confusing.  In the abstract, the submission promised a "a comprehensive analysis of both the sharpening phase and the EoS regime throughout the entire GD dynamics, focusing on shallow ReLU networks with squared loss on orthogonal inputs."  However, the paper then proceeded to ... not do this.  The submission does not contain a comprehensive analysis of the progressive sharpening and EoS regime.  The main claimed theoretical result seems to be Theorem 3.1.  However, the paper does not actually include a proof of this theorem (?!), the quantities in the the theorem statement are not defined in the paper (what are the phases?  what is $\lambda$?  what are $r$ and $\tau$ and $\epsilon$?), and, finally, the theorem does not even seem to provide a comprehensive analysis of EoS, just a lower bound on the growth in the eigenvalues (the progressive sharpening), but we don't know how tight this bound is.  The theorem says nothing about EoS.

All of the experiments are just reproductions of analogous experiments from Cohen et al (2021), and are in no way explained by the the theory in the paper.

**Strengths:**

see summary

**Weaknesses:**

see summary

**Questions:**

Is my characterization of the paper correct?

---

### Official Review · Reviewer_QK5F · 2023-11-07

**Soundness:** 1 poor
**Presentation:** 2 fair
**Contribution:** 2 fair
**Rating:** 1
**Confidence:** 3

**Summary:**

This paper aims to analyze the progressive sharpening (PS) phenomenon and the edge of stability (EoS) for the GD training dynamic of two-layer ReLU neural networks with square loss on orthogonal inputs. The authors showed that both PS and EoS can appear for multiple top eigenspaces of the Hessian matrix during the GD training process.

**Strengths:**

This paper considered a more general regime where all the layers of the two-layer neural networks are trained simultaneously by GD. The authors introduced advanced techniques from Boursier et al. (2022) to analyze PS and EoS.

**Weaknesses:**

The paper is not complete and there are places that need to be polished. Specifically, the notations and mathematical formulas in Section 2.2 and Section 3 are not fully explained. There are lots of typos in these sections and the appendix which hinder readers from understanding the theoretical results of this paper. It is hard to understand Theorem 3.1 and conclude the final claim that the authors made in the introduction and conclusion. There is no complete proof of Theorem 3.1 and detailed explanations or remarks on Theorem 3.1. Besides, the authors should summarize the innovation points of this paper, the methods and results borrowed by Boursier et al. (2022), and how the current paper differs from Boursier et al. (2022).

**Questions:**

1. In Figure 1, can you also plot the dynamic of the test losses during training? This may help us understand the relationship between the generalization and EoS, like Ahn et al. (2022) in the reference.

2. Any other references that used orthogonal or nearly orthogonal datasets in deep learning theory? It would be better to mention more references with this Assumption 2.1.

3. How do you derive Eq. (3) from Eq. (2)?

4. In Lemma 2.1, what is $\mathbf{d}_j^T$? What does $T$ stand for in this lemma?

5. In the Assumption 2.2, what is $\mathbf{w}_j^k$? Here $k\in [n]$, right? Can you give an example of when Assumption 2.2 is satisfied?

6. In Theorem 3.1, what are $\mathbf{d}+$ and $\mathbf{d}_{-}$? And there are typos in the definitions of  $\lambda +$ and $\lambda -$. In this theorem, I did not see how the result rely on the number of iterations. How do we know in which iteration of GD, the neural network goes into the EoS phase?